# Study protocol: Examining sexual and reproductive health literacy in Mexican American young women using a positive deviance approach

**Lindsay M. Batek** [1] *, **Natalie M. Leblanc**[1], **Amina P. Alio**[2], **James M. McMahon**[1]

1 School of Nursing, University of Rochester, Rochester, NY, United States of America, 2 University of Rochester Medical Center, Rochester, NY, United States of America

\* lindsay_batek@URMC.rochester.edu

**Data Availability Statement:** This manuscript presents a study protocol, and the research is actively ongoing, and data collection and analysis

## Abstract

Health literacy is generally low in marginalized groups, leading to delays in accessing care, poor health outcomes, and health disparities. Yet, some individuals in these groups demonstrate higher health literacy and better health outcomes. These exceptional cases exemplify 'positive deviance' because they have found ways to be successful where others have not. Identifying the methods, practices, and resources that these individuals have used to gain health literacy and healthcare access may have generalized application to improve health literacy, access to care, and health outcomes. Using the Integrated Model of Health Literacy, the main objectives of this study are to (1) identify facilitators, barriers, and strategies to gain sexual and reproductive health literacy and healthcare access and (2) to explore each of the core domains of health literacy as they relate to successful access of sexual and reproductive healthcare services among individuals identified as positive deviants. For the purposes of this mixed methods community engaged study, positive deviants are defined as Mexican American young women aged 18–29 years old living in Rural Western New York who have accessed sexual and reproductive healthcare within the past year. A community advisory committee will be formed to provide community-engaged guidance and support for the recruitment of participants. Positive deviants will participate in a survey and semi-structured interview. Data collection and analysis will be simultaneous and iterative. Results will provide evidence of positive deviant methods, practices, and strategies to gain health literacy and access to sexual and reproductive healthcare. Findings may reveal characteristics and patterns in the relationship of health literacy and healthcare access that can inform interventions to improve health literacy and make healthcare more accessible for this demographic group and context.

## Introduction

Health literacy is a social determinant of health and low health literacy has a strong association with health outcomes, disparities, and equity. Health literacy is the degree to which individuals

are not yet complete. Consequently, there are no final datasets available to share at this stage.

**Funding:** This study was supported by a grant to LB by the University of Rochester CTSA (UL1 TR000042) from the National Center for Advancing Translational Sciences of the National Institutes of Health. (https://www.urmc.rochester.edu/clinical-translational-science-institute.aspx).

**Competing interests:** The authors have declared that no competing interests exist.

"have the ability to find, understand and use information and services to inform health-related decisions. . ." [1]. Despite the generalization that health literacy is low within marginalized groups, there is a range in the level of health literacy for individuals within these groups. Moreover, given that some individuals within marginalized groups have higher health literacy, methods to identify and promote the behaviors and practices that have demonstrated effectiveness in their ability to gain health literacy may have generalized application to increase the efficiency and effectiveness of targeted interventions.

One method that examines uncommon but successful health promotion behaviors adopted by certain individuals within a marginalized group is termed "positive deviance" [2]. Positive deviance is an asset-based approach to healthcare research that builds upon strengths, resources, and successful strategies and behaviors utilized by members within a community to discern what is working and why, and what can be learned by these successes [3]. Once these assets are identified, they can be scaled up to improve outcomes for others experiencing the same challenges. Asset-based approaches contrast with deficit-based approaches which may lead to outcomes that do not last or fail to reach the most disadvantaged and may have unintended consequences [3]. One key aspect of positive deviance that differentiates it from other research methods is that solutions to challenges are generated from within the community. This method supports the feasibility of using available resources and enhances uptake of health promoting behavior and their sustainability over time [3]. Examining the strategies and behaviors of those exhibiting exceptional performance, or atypical ("deviant") positive behaviors can provide valuable insights to inform interventions to improve community-level outcomes overall. Viewing health literacy and healthcare use with a positive deviance lens has the potential to be a powerful strategy to substantively decrease health disparities.

This project will explore how some Rural Mexican American Young Women (RMAYW) have gained sexual and reproductive health literacy and access to sexual and reproductive healthcare in Rural Western New York. RMAYW are a group with longstanding sexual and reproductive health disparities including high teen pregnancy rates, higher incidence of sexually transmitted infections (STIs), decreased or delayed use of prenatal care [4, 5] and decreased rate of Pap screening [6]. While statistics regarding the health literacy of this specific group are unavailable, existing evidence suggests that health literacy is generally low among marginalized and socioeconomically disadvantaged populations [7]. This evidence strongly indicates that health literacy levels are likely low among RMAYW, a group often situated within these marginalized and lower income brackets. RMAYW have multiple discrete characteristics that are each associated with lower health literacy including, being Hispanic, living in rural areas, and frequently have an uninsured or underinsured health insurance status [8]. Health literacy is lower for those who were raised speaking Spanish or who are monolingual Spanish speakers [9]. In these rural areas, not only is there a scarcity of healthcare providers, but there is also a notable absence of cultural and ethnic diversity among those available to deliver care [8]. Yet, a subset of RMAYW successfully engage in healthcare, enabling them to avail individualized guidance and care. For the purposes of this study, successful engagement of healthcare means participants have received care for sexual or reproductive healthcare within the past year. Healthcare providers provide access to the most effective forms of contraception, preventative measures, prompt treatment for STIs, and timely prenatal care. While barriers and facilitators to healthcare have been recognized, and are indeed helpful, they do not fully explain the methods by which RMAYW utilize facilitators or surmount barriers to obtain care. These mechanisms, strategies, and pathways they use to access care that supports their sexual and reproductive health warrants exploration.

Previously reported barriers to healthcare service use by Hispanic women, including Mexican American women, included beliefs and misconceptions about contraception or fertility

[10–12], lack of risk appreciation [10, 13], lack of knowledge [11, 13–18], perceptions of healthcare provider discrimination [10, 11, 14], language barriers, poor communication [14], a need for interpreters [11], as well as logistical issues such as lack of transportation [11, 12], lack of insurance, cost [11, 16, 18], and lack of continuity of care [18]. Facilitators to care identified in previous research for Hispanic women include clear future plans [12], higher education [19], clarity of intention to avoid pregnancy [14], familial communication and support [20], partner or peer support [14, 20], and good communication and trust with healthcare providers [12, 14]. These previously identified barriers and facilitators align with the personal, situational, societal, and environmental determinants in the Integrated Health Literacy Model [21]. These determinants form the backdrop against which individuals enact the fundamental literacy constructs of accessing, understanding, appraising, and applying health information [21]. As depicted by the circular and bidirectional arrows in the model (Fig 1), these determinants and core constructs serve as both antecedents and consequences of health literacy, contributing to the dynamic process of ongoing health literacy changes [21].

Health literacy, as defined by Sørensen et al. (p. 3) [21], "is linked to literacy and entails people's knowledge, motivation and competences to access, understand, appraise, and apply health information in order to make judgments and take actions in everyday life concerning healthcare, disease prevention and health promotion to maintain or improve quality of life during the life course." Additionally, the Integrated Health Literacy Model [21], includes contextual influences on health literacy including societal, environmental, structural, and personal determinants. Fig 1 illustrates how the Integrated Health Literacy [21] is being conceptualized for application in this project. Understanding how these core competencies (knowledge, motivation, competence, and determinants) interact to affect health literacy and healthcare service use in those exhibiting positive deviant behaviors can reveal specific strengths and successful strategies and pathways for RMAYW and potentially other marginalized groups.

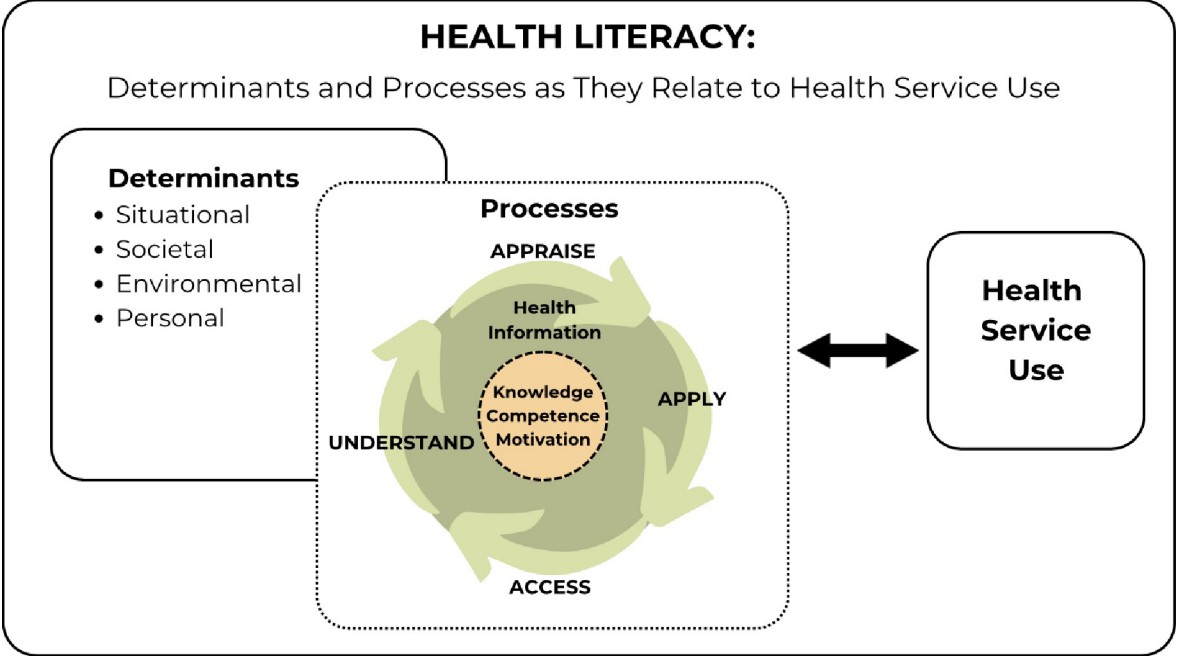

**Fig 1. Conceptualization of model of integrated health literacy [21].** This Integrated Model of Health Literacy [21] is conceptualized as depicted to for application in this study.

This study has the following aims:

**Aim 1**: In participants identified as 'positive deviants,' explore each of the core domains that comprise health literacy as they relate to successful access to sexual and reproductive health-care services. The domains include knowledge, competence, motivation, access, understanding, appraisal, and application of health information and societal, situational, environmental, and personal determinants.

**Aim 2:** In participants identified as 'positive deviants,' identify strategies they have used to attain sexual and reproductive health literacy and access to sexual and reproductive healthcare.

**Aim 3:** Identify 'positive deviant' recommendations to promote health literacy and health care access and use for Mexican American women living in the U.S.

Accomplishing these specific aims will reveal effective strategies used by RMAYW to navigate barriers to sexual and reproductive health literacy and healthcare. Findings will inform how health literacy can enhance translational science research using a positive deviance approach to identify strategies that are already successfully in use (for some) and that may be generalizable and sustainable and able to fulfill unmet needs on a larger scale.

## Methods

### Study design

This community engaged research will use a mixed method (QUAL+quan) design. The purpose of the design is to enable the use of complementarity, completeness, expansion, triangulation and diversity to holistically explore these aims [22]. This study uses a multiple paradigm stance with the assumptions of pragmatism and critical realism and uses a concurrent design [22]. The qualitative part of the study will use a qualitative descriptive design [23, 24] and the quantitative part will use measures of health literacy, religiosity and acculturation to investigate RMAYW's sexual and reproductive health literacy, as well as the resources and paths they have taken to access sexual and reproductive healthcare. Formation of a Community Advisory Committee comprised of healthcare providers who care for RMAYW, community health workers and community members will guide the research process and provide diverse perspectives on sexual and reproductive healthcare for RMAYW. The sample of study participants will include women between18-29 years of age who identify as Mexican or Mexican American, many of whom may be from migrant communities, living in Rural Western New York and who have accessed sexual and reproductive healthcare within the past year. The structure of this community engaged study is illustrated on Fig 2.

To accomplish the objective of exploring the domains of health literacy, an open-ended platform will be used through which acquisition of sexual and reproductive health literacy can be explored individually with RMAYW as well as through separate group discussions with the Community Advisory Committee. As interviews are completed and iterative data collection and analysis ensue, findings—especially those pertinent to potential interventions—will be reviewed with the Community Advisory Committee so that results are informed by both users as well as providers, community health workers, and community members.

### Community Advisory Committee

A Community Advisory Committee will be formed during the recruitment phase of this project. It will consist of medical professionals and community health workers who provide sexual and reproductive healthcare to RMAYW, as well as members of the Rural Western New York

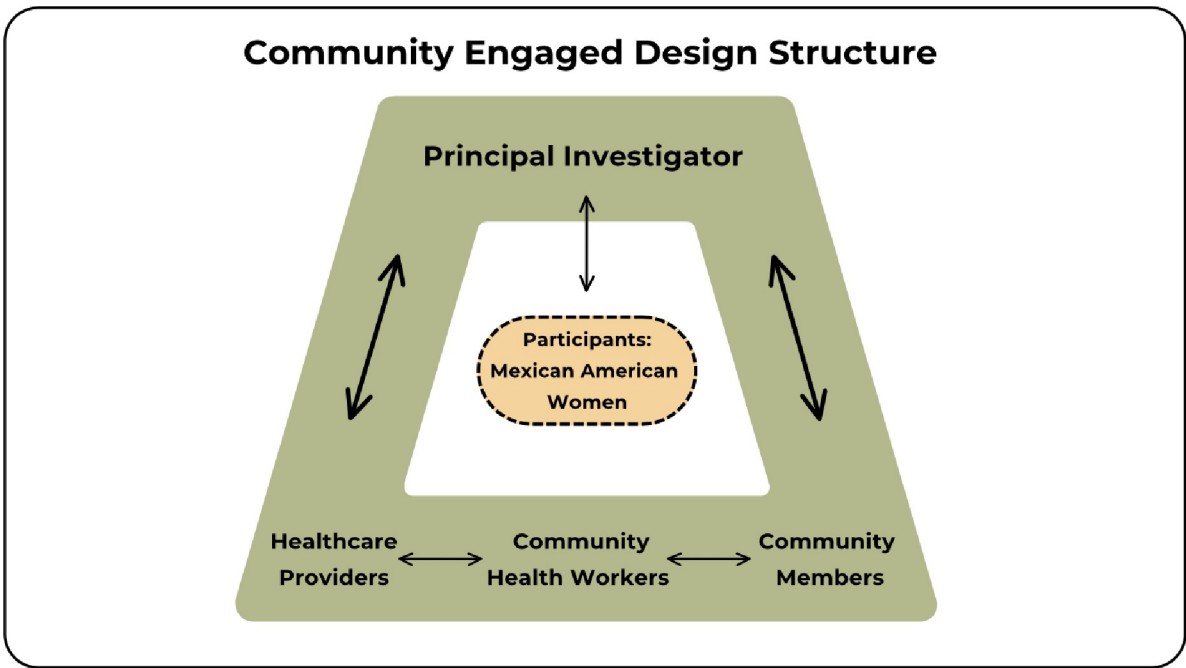

**Fig 2. Community engaged design structure.** The Community Advisory Committee is comprised of the PI, healthcare providers, community health workers and community members to provide guidance and insights into the study design, recruitment, and data collection.

Mexican American community. Perspectives from the Community Advisory Committee will provide key informant points of view and enhance the quality and appropriateness of the research regarding health literacy and sexual and reproductive healthcare access for RMAYW [25]. The Community Advisory Committee will offer additional data and guide the research processes of instrument development (interview guide), recruitment, and cultural interpretation. They will not have direct contact with the subjects during the study and will not have access to identifiable data collected during the study. During the final analysis phase of the project, they will provide feedback on study findings regarding suggestions and recommendations made by the participants to improve health literacy and healthcare access. The estimated size of the Community Advisory Committee will be eight to ten individuals, who will meet quarterly during this project with additional contact as needed. Retention strategies for the Community Advisory Committee members include a monetary honorarium ($100) for each attended meeting.

### Community Advisory Committee inclusion criteria and recruitment

Recruitment criteria for Community Advisory Committee members who are community health workers and healthcare providers are those who facilitate and provide care for RMAYW at participating clinics (medical doctors, midwives, nurse practitioners or physician assistants). Recruitment criteria for community members will be those who serve as resources to RMAYW who are seeking healthcare, such as leaders in the *Mujeres Divinas*, a local organization founded by migrant worker women to support migrant women. Community Advisory Committee inclusion criteria for health care providers and community health workers is that they must support or facilitate Mexican or Mexican American women using healthcare services or provide care for them and they must be willing to serve as study consultants throughout the anticipated one-year duration of the project.

The Community Advisory Committee will be recruited via study personnel by contacting the local community health clinic administrators as well as by word of mouth within the University of Rochester Medical Center healthcare provider community. The study personnel will interview potential candidates and assess for eligibility to join the Community Advisory Committee. If eligibility is met and the individual is interested and willing to participate, they will be invited to join the Community Advisory Committee.

## Study participants and eligibility

The study uses a concurrent, purposive and convenience sampling design [22] to focus on a specific demographic: females (self-identified as assigned female sex at birth) of Mexican origin (either born in Mexican or in the U.S.), who are 18–29 years of age, can read and speak English or Spanish, and who have used sexual and reproductive healthcare services within the past year (such as contraception or STI counseling, prevention or treatment, pregnancy related care, Pap screening or annual gynecological well woman care). Some members of this community are a vulnerable population due to economic disadvantage and/or due to undocumented immigration status. Because of this, precautions will be taken not to obtain any personal identifying information and consent and verification of receipt of financial incentive will be verbal. Participants will be from Monroe, Wayne, Ontario, Orleans, Genesee, or Livingston Counties in Western New York and must have the ability to be present for an interview in person, by phone or video conferencing at a mutually selected time. Study materials and spoken interactions will be provided in either English or Spanish, according to the participant's preference. The total sample size is anticipated to be between 15–25. This sample size is an estimation based on when data saturation has been reached in previous studies [26, 27]. Enrollment will continue until data saturation is achieved and additional interviews fail to provide novel themes. If participants withdraw before completion of the interview they will be replaced until data saturation is reached, if they choose to withdraw at end of the interview, they may choose to have the content of their interview and/or survey redacted. Participants may be receiving care at any of the clinics that are participating in the study. Pregnant women may participate in the study if they meet the eligibility criteria, although the study does not specifically target this demographic.

## Ethics approval

This study has been approved by the Institutional Review Board at the University of Rochester (STUDY00008311).

## Recruitment

Women living in rural areas in Western New York State who identify as Mexican or Mexican American will be recruited from various healthcare clinics that provide sexual and reproductive healthcare. Potential research sites Finger Lakes Community Health Care Center Network (such as Geneva, Penn Yan, Sodus, and Newark) and Oak Orchard Health clinic sites (such as those in Brockport and Batavia). This method of recruitment automatically identifies women using "positive deviant strategies" in that they are currently receiving sexual and reproductive healthcare.

Recruitment materials will be co-created with and provided to the Community Advisory Committee members who are health care providers so they can be distributed to the clinics where they work. Recruitment materials, including flyers and posters, will be displayed in the waiting areas and clinic rooms of participating clinics, and staff will also distribute them to ensure visibility to potential participants. Extra flyers will be available at the front desk or from

providers so the potential participant can take a flyer with her. Flyers will be shared with local organizations to promote study recruitment, and they may disseminate them electronically through their networks. Either by self-identification via the informational materials or through referrals by clinic staff or provider members of the Community Advisory Committee, potential participants can explore the opportunity to enroll in the study. Flyers will list a phone number and email with instructions to contact study personnel if interested in participating or to learn more. Flyers will also have a QR code that links to a REDCap pre-screening eligibility survey to determine if a potential participant meets inclusion criteria.

## Eligibility screening and enrollment

We have developed three paths to screening to accommodate the needs and preferences of prospective participants: by phone, email and QR code, all found on the flyer. Subjects who call on the phone can be screened by phone if they prefer, or they can be sent an email that contains a link to the REDCap screening form, or those individuals can be screened by phone if they wish or if they prefer not to share an email address. If screened via phone, an Information Sheet will be emailed or mailed to the participants or read to them and then given a hard copy in person if they choose not to share email or mailing address. If they access the REDCap link via the email link or QR code, they can answer the eligibility questions directly via computer or mobile device. In all cases, if eligible, the Information Sheet will be provided by email or on paper or verbally read to potential subjects and subjects can document whether they want to participate, and if so, contact information is collected. If eligibility screening takes place in person and if the interview is to be conducted at that time, contact information will be unnecessary and will not be collected. If eligibility screening occurs on REDCap, contact information (name, email, and phone number) will be collected to arrange a meeting time for an interview. If potential participants are screened as ineligible they will be thanked for their time, and have their screened variables collected and documented in REDCap to record reasons for ineligibility. All the screening information collected on REDCap will be stored in an approved secure REDCap database, which include the following information: name, date of contact, ways of learning about the study, result of each eligibility criterion.

## Data collection

Upon enrollment (after the eligibility screening and review of the Information Sheet with the opportunity to ask questions and receive answers), an in-person, phone or videoconference meeting will be scheduled with the participant. Once the meeting has begun, REDCap will be used to administer the questionnaire and then the semi-structured interview will be conducted. The meeting may last up to 90 minutes. Interviews will be audio-recorded following Health Insurance Portability and Accountability Act compliant procedures and subsequently saved on a secure institutional server. After these recordings are transcribed and the transcriptions verified for accuracy, the original audio files will be erased from the recording devices. During the transcription process, any identifying information will be removed so that the transcripts are de-identified. To minimize risk, an optional voice changing platform will be offered, as an additional layer of protection to participants as a consideration for those who may be undocumented immigrants. In this case, the original audio recording of the interview will be re-recorded using the voice changing platform and stored and used for analysis in the altered form. Once the quality of the voice altered recording is deemed adequate, the original recording will be deleted. Identifying data will be kept separately from research data collected during the interviews and linked only by an ID number. At the close of the interview, the participants

**Table 1. Data collection description.**

| | Content Description |
|---|---|
| **Survey Questionnaire** | Demographic Information |
| | Brief Sexual and Reproductive Health History |
| | Acculturation Measure: BASH |
| | Religiosity Questions |
| | Health Literacy Questions |
| **Semi-Structured Individual Interview** | Description of barriers and facilitators to information and healthcare related to sexual and reproductive healthcare. |
| | Description of strategies that you have used to sexual and reproductive health information and healthcare. |
| | Knowledge, motivation, and competence with respect to sexual and reproductive health information and healthcare services. |
| | Accessing of sexual and reproductive health information and healthcare services. |
| | Understanding of sexual and reproductive health information and healthcare services. |
| | Appraising of sexual and reproductive health information and healthcare services. |
| | Application of sexual and reproductive health information and healthcare services. |
| **Close of Interview** | Opportunity for subject to make suggestions about how to facilitate health information and healthcare use for herself and her peers. |
| | Debrief and clarify any points in the interview that may need to be revisited for clarity. |

will receive cash or a gift card (if in-person) or if via zoom, the gift card will be mailed to the subject at the address she provided.

The data that will be collected during one meeting of up to 90 minutes in length is summarized in Table 1.

Table 1: Data Collection Description. Data collection will include survey data and interview data using the content as described.

## Quantitative study measures

The quantitative component of this study will consist of a REDCap-administered survey that will include demographic characteristic, as well as scale instruments measuring acculturation, religiosity, and health literacy.

*Acculturation* is measured using the Brief Acculturation Scale for Hispanics (BASH) [28]. The BASH asks the following four questions: what language do you speak at home, what language do you speak with your friends, what language do you read in, what language do you think in with the six optional answers: only Spanish, Spanish more than English, Spanish and English equally, English more than Spanish, only English, Language other than English or Spanish [28]. The BASH has high reliability ($\alpha$ = .92) and validity among Mexican American young adults with respect to generation ($r$ = .74), length of time in the US ($r$ = .59), and subjective measure of acculturation ($r$ = -.43) [28]. Higher acculturation scores were significantly associated with being born in the United States as well as for those who chose English as their language of preference for the survey [28].

Religiosity will be measured with 3 questions from the Centrality of Religiosity Scale (CRS-5) which consists of 5 items that measure the different aspects of one's religiosity including intellect, ideology, public practice, private practices, and religious experience. Two questions were omitted from the 5-question scale based on preliminary feedback from the Community Advisory Committee that questions #2 and #5 appeared inappropriate for this population. The

scoring of one's responses differentiates three groups: very religious individuals, religious individuals, and non-religious individuals. This scale has been found to have a Cronbach's alpha (α) coefficient of 0.85 [29, 30]. The CRS-5 has been translated to Spanish by study personnel for use in this study. The principal investigator will initially translate the questions from English to Spanish, and then the bilingual Community Advisory Committee members will review the translation for accuracy and modify as needed.

*Health Literacy* is measured by 12 questions inspired by the Integrated Health Literacy Model [21] and HLS-EU-Q [31] about the participant's experience accessing, understanding, appraising, and applying sexual and reproductive health information and accessing and using sexual and reproductive healthcare. For example, one questions asks participants to choose how easy or difficult they would say it is for them to access or find the information she needs to make decisions regarding her sexual and reproductive health with responses being a Likert-type scale with four choices "very easy, easy, difficult or very difficult." These answers to these questions, combined with interview findings will fulfill the objectives of Aim 1.

In addition, a single question will assess how confident the participant is at filling out medical forms by themselves, with the following response options: extremely confident; quite confident; somewhat confident, a little confident, not at all confident [32–34]. In three studies that used this question had summarized likelihood ratios of 5.0 (95% CI, 3.8–6.4) for the two least confident groups (a little or no confidence) of LR of 2.2 (95% CI, 1.5–3.3) for somewhat confident of 2.2 (95%, CI,1.5–3.3), and for the quite a bit or extremely confident LR of 0.44 (95% CI, 0.24–0.82) [35]. This measure of health literacy has also been validated in Spanish [36].

## Data analysis

Qualitative data collection and analysis will be a simultaneous and iterative process. Member checking at the end of each interview will ensure trustworthiness of the data. During this process the researcher will share her understanding and interpretation with the participants and provide the opportunity for the participant to clarify or correct what was understood from the interview [37]. Interviews will be transcribed verbatim (in English or Spanish) and translated to English (if necessary) by the person who conducted the interview (study personnel). The accuracy of the transcribed interviews will be confirmed by listening to the audio recordings and confirming accuracy of each word in the transcription and translation (for Spanish interviews). All personal identifying information will be deleted from the transcript. A participant ID will be assigned to each interview/subject will be used to maintain confidentiality. The transcripts without personal identifiers will be used for analysis. Dedoose software (Dedoose.com) will be used for data management and analysis. The transcripts will be read, and the content will be inductively analyzed and coded to develop initial themes. Inductive and deductive theoretical reasoning will be used [22]. An audit trail will be developed to keep a record of decisions guiding analysis. A codebook will be developed iteratively as the data collection and analysis process continues. The results of the quantitative measures will allow description of the sample and for organization and grouping of the analysis to make comparisons of the qualitative data by the health literacy level indicated by the single question measure. Data collected from the quantitative surveys and qualitative interviews will be analyzed to fulfill objectives for Aims 1 (health literacy characteristics of participants) and 2 (strategies used by participants to access healthcare). As the analysis is finalized, the results will be presented to the Community Advisory Committee to obtain feedback and closing perspectives from the group. Recommendations made by participants will be discussed at the final Community Advisory Committee meeting, the results of which will fulfill the objective of Aim 3.

## Status and timeline

The estimated timeline for this study is completion within one year.

## Discussion

Health literacy is the degree to which individuals have the ability to find, comprehend, and utilize information and services to improve their health [1]. Lower health literacy is associated with suboptimal insurance coverage, delays in care or forgone care and higher disease burden [38]. Individuals with lower education, racial and ethnic minoritized groups, lower health status or who are older are more likely to have lower health literacy [38], thus contributing to significant health disparities in these groups. Most commonly, research regarding health literacy uses a deficit approach identifying the reasons or contributors to delays or forgone care and removing these barriers. While this may provide insight, it does not capture what compels one to move forward, or what motivates and supports an individual's path to care. This proposal uses a strength-based or "positive deviance" approach to understand how some from a group with traditionally lower health literacy and healthcare access have successfully accessed care. By examining user assets, motivations, and innovations to access healthcare, specific action-promoting aspects of health literacy can be identified and magnified in implementation and translational science interventions. This is particularly important to achieve health equity in persistently marginalized groups, and as a way to address lower levels of health literacy, healthcare access and disproportionately poor outcomes.

Application of a positive deviance approach to contextualize the relationship between sexual and reproductive health, health literacy and care access in rural Mexican American young women can reduce health disparities through translational science by yielding interventions that are feasible (using available resources), functional (with proven success), and culturally acceptable (sourced from others within same cultural group). The proposed study uses this methodology to address this gap by both seeking to understand aspects of health literacy that serve not only as facilitators but also motivators of action in rural Mexican American young women to seek and access care as well as the resources used in doing so.

Findings from this study can be used to inform interventions to improve health literacy, and healthcare access and use in this population. Future studies can implement and test interventions that decrease delays in care and disease burden through improved access and use of healthcare. Addressing health disparities in this way is appealing to funding sources because the methods used are pragmatically and closely tied to perspectives and experiences of success from within marginalized groups. As sexual and reproductive healthcare disparities for rural Mexican American young women are addressed with interventions stemming from this work, the relationship between health literacy, positive deviance and translational science modeling would continue to advance, streamlining this approach for future research.

## Author Contributions

**Conceptualization:** Lindsay M. Batek, Amina P. Alio, James M. McMahon.

**Funding acquisition:** Lindsay M. Batek, James M. McMahon.

**Methodology:** Lindsay M. Batek, Natalie M. Leblanc, Amina P. Alio, James M. McMahon.

**Project administration:** Lindsay M. Batek, Natalie M. Leblanc, James M. McMahon.

**Resources:** Lindsay M. Batek, Amina P. Alio, James M. McMahon.

**Supervision:** Natalie M. Leblanc, Amina P. Alio, James M. McMahon.

**Visualization:** Lindsay M. Batek.

**Writing – original draft:** Lindsay M. Batek.

**Writing – review & editing:** Lindsay M. Batek, Natalie M. Leblanc, Amina P. Alio, James M. McMahon.

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
