## [Decision Letter · Decision Letter 0]

27 Mar 2024

PONE-D-23-42586Study protocol: Examining sexual and reproductive health literacy in Mexican American young women using a positive deviance approachPLOS ONE

Dear Dr. Batek,

Thank you for submitting your manuscript to PLOS ONE. After careful consideration, we feel that it has merit but does not fully meet PLOS ONE’s publication criteria as it currently stands. Therefore, we invite you to submit a revised version of the manuscript that addresses the points raised during the review process.

Please include the following items when submitting your revised manuscript:A rebuttal letter that responds to each point raised by the academic editor and reviewer(s). You should upload this letter as a separate file labeled 'Response to Reviewers'.A marked-up copy of your manuscript that highlights changes made to the original version. You should upload this as a separate file labeled 'Revised Manuscript with Track Changes'.An unmarked version of your revised paper without tracked changes. You should upload this as a separate file labeled 'Manuscript'.If applicable, we recommend that you deposit your laboratory protocols in protocols.io to enhance the reproducibility of your results. Protocols.io assigns your protocol its own identifier (DOI) so that it can be cited independently in the future. For instructions see: https://journals.plos.org/plosone/s/submission-guidelines#loc-laboratory-protocols. Additionally, PLOS ONE offers an option for publishing peer-reviewed Lab Protocol articles, which describe protocols hosted on protocols.io. Read more information on sharing protocols at https://plos.org/protocols?utm_medium=editorial-email&utm_source=authorletters&utm_campaign=protocols.

We look forward to receiving your revised manuscript.

Kind regards,

Mary Diane Clark, PhD

Academic Editor

PLOS ONE

 [This study was supported by a grant to LB by the University of Rochester CTSA (UL1 TR000042) from the National Center for Advancing Translational Sciences of the National Institutes of Health. (https://www.urmc.rochester.edu/clinical-translational-science-institute.aspx).].  

[This study was supported by a grant to Dr. Batek by the University of Rochester CTSA (UL1 TR000042) from the National Center for Advancing Translational Sciences of the National Institutes of Health.]

  [This study was supported by a grant to LB by the University of Rochester CTSA (UL1 TR000042) from the National Center for Advancing Translational Sciences of the National Institutes of Health. (https://www.urmc.rochester.edu/clinical-translational-science-institute.aspx).].

Additional Editor Comments:

Thank you for allowing us to review you piece. It is critical and the positive framework is nice to see utilized.

I have a few comments here

Can we avoid abbreviations? PD and RMAYW then SRHL next SRHC. And then later CAC (for this one -- you introduce it and then don’t use it in the next time)

APA suggest using only that that are common knowledge to those outside of your area

So STI works as that is a known abbreviation

I also have others directly on the manuscript.

The reviewers also made some comments---many of them overlap with redundancy.

Reviewers' comments:

Reviewer's Responses to Questions

**Comments to the Author**

1. Does the manuscript provide a valid rationale for the proposed study, with clearly identified and justified research questions?

Reviewer #1: Yes

Reviewer #2: Yes

2. Is the protocol technically sound and planned in a manner that will lead to a meaningful outcome and allow testing the stated hypotheses?

Reviewer #1: Yes

Reviewer #2: Yes

3. Is the methodology feasible and described in sufficient detail to allow the work to be replicable?

Reviewer #1: Yes

Reviewer #2: Yes

4. Have the authors described where all data underlying the findings will be made available when the study is complete?

Reviewer #1: No

Reviewer #2: No

5. Is the manuscript presented in an intelligible fashion and written in standard English?

Reviewer #1: Yes

Reviewer #2: Yes

6. Review Comments to the Author

You may also provide optional suggestions and comments to authors that they might find helpful in planning their study.

Reviewer #1: Dear authors,

Thank you for your efforts in studying marginalized groups and individuals who speak languages other than English, such as Spanish. There is still much more work to be done beyond solely including English speakers. Your efforts are appreciated.

Every research endeavor demands a significant investment of effort; therefore, maximizing its validity whenever possible is essential. Two key suggestions to enhance validity include referencing the (1) mixed-method research protocol outlined by Venkatesh, V., Brown, S.A., and Sullivan, Y.W. (2016) in the "Guidelines for Conducting Mixed-methods Research: An Extension and Illustration," Journal of the AIS (17:7), 435-495, and (2) utilizing the forward-backward translation technique proposed by Guillemin F, Bombardier C, and Beaton D. in "Crosscultural adaptation of health-related quality of life measures: literature review and proposed guidelines," J Clin Epidemiol. 1993;46:1417-1432.

Other comments are as follows:

1. In lines 36-38, to clarify the role of the Community Advisory Committee (CAC) in this study, it may be beneficial to explicitly state that the CAC will be formed to provide community-engaged guidance and support for the recruitment of participants.

2. In line 87, does "successful" refer to the criteria detailed in lines 212-215? If so, it would enhance clarity for readers to provide examples of what is meant by "success" immediately. This is important as "success" is one of your main definitions.

3. Line 90-94: Are these two sentences the same?

4. In line 128-132, according to Figure 1, "environmental" is one of the variables, but it is omitted in Aim 1. Is this intentional, and if so, why?

5. In line 143, to ensure the validity of the mixed method protocol, please consider incorporating the mixed-method protocol suggested by Venkatesh, V., Brown, S.A., and Sullivan, Y.W. (2016) in the "Guidelines for Conducting Mixed-methods Research: An Extension and Illustration," Journal of the AIS (17:7), 435-495. For instance, questions regarding the property of mixed-method research, such as whether quantitative and qualitative data collection occur sequentially or concurrently, are not explicitly addressed.

6. In line 161-164, according to Figure 1, "environmental" is one of the variables, but it is omitted in Aim 1. Is this intentional, and if so, why?

7. In line 188, are "medical providers" and "healthcare providers" referring to the same group of individuals? This question arises because in Figure 2, "medical providers" are not included.

8. In line 264, could you please clarify what occurs if participants choose to withdraw after completing the interview?

Line 269: What is HIPAA? Do you mean the Health Insurance Portability and Accountability Act? Please clarify the full term for readers who may not be familiar with the abbreviation.

9. Line 295: Is it 5 or 6 options? Please double-check.

10. In line 302, why are only 3 questions out of the 5 items from the Centrality of Religiosity Scale (CRS-5) being used to measure religiosity? Please clarify

11. In line 307, could you explain how the validity of the translation is ensured, particularly considering the utilization of forward and back translation techniques?

12. In lines 328 and 329, please consider including forward and back translation techniques to ensure validity.

Line 364: "RYMAW," is this a typo?

13. Please clarify if the data collected would be share with the journal or not

14. Please check the labeling of Fig 2.

Reviewer #2: Thank you very much for the opportunity to review this protocol. I found it to be a well-written article that provides a helpful description of both the rationale and planned steps for the research to be conducted. My suggestions below are mostly to clarify several pieces of information articulated in the protocol to enhance the consistency with which similar concepts are portrayed throughout the article and to further distinguish between concepts that would benefit from clearer separation from one another.

1. Introduction section, 3rd paragraph, 3rd to last and 2nd to last sentences: It was not clear what distinct information is being conveyed by these two sentences (one starting with “While barriers and facilitators to healthcare …” and the other with “While barriers and facilitators to care …”). Please consider either combining them or further differentiating their content from one another.

2. Near the end of the Introduction section, where the three specific aims are mentioned, please consider sharpening the distinction between Specific Aims 2 and 3.

3. Throughout the Methods section, please consider making clearer the exact study tasks to be carried out for each specific aim.

4. The Study Design section’s 5th sentence (which starts with “National data indicate low health literacy … for these women …”) seems to contradict the Introduction section’s 3rd paragraph’s 3rd sentence, which states that “statistics regarding the health literacy of this specific group are unavailable.” Please consider resolving this seeming discrepancy.

5. The Study Participants and Eligibility section’s 3rd sentence states that “precautions will be taken not to obtain any personal identifying information,” while the Eligibility Screening and Enrollment section’s last sentence mentions the storage of identifying information. Please consider clarifying how these planned tasks are consistent.

6. In the Recruitment section, please consider elaborating on whether the study information disseminated through the flyers and posters will be made available to potential participants electronically/virtually as well (and if so, how), in addition to being displayed and distributed at clinic locations and extra flyers being available for potential participants to physically pick up.

7. Data Collection section: Please consider providing a justification for why the extra layer of protection using the voice changing platform is by choice of the participant and only offered to some participants, rather than being uniformly applied to all participants.

8. Regarding data availability, this submission is accompanied by the statement “All relevant data from this study will be made available upon study completion.” It is unclear how this will be done and what steps will be taken to protect participant confidentiality in doing so, especially given the potentially sensitive topics covered by the semi-structured interviews.

7. PLOS authors have the option to publish the peer review history of their article (what does this mean?). If published, this will include your full peer review and any attached files.

Reviewer #1: No

Reviewer #2: **Yes: **Bo Kim

---

## [Author Response · Author response to Decision Letter 0]

26 Apr 2024

April 26, 2024

PONE-D-23-42586

Study protocol: Examining sexual and reproductive health literacy in Mexican American young women using a positive deviance approach

Dear Dr. Mary Diane Clark and reviewers, 

Thank you for reviewing my manuscript and providing constructive critiques and suggestions to improve it. In the following pages, I have responded to each of comments, questions, and suggestions about the manuscript. I have structured my response to each point using the order from the editor and reviewers’ letters. In addition to this rebuttal letter, I have submitted a marked-up copy of the manuscript labeled ‘Revised Manuscript with Track Changes’ and an unmarked version of the revised paper without tracked changes labeled ‘Manuscript.’

I appreciate your general and specific comments about redundancy, I have tried to implement your suggestions and I believe that redundancy is reduced in this revised manuscript. The details of the changes are available below and also visible on the manuscript with track changes. 

I have updated my financial disclosure statement; the updated statement follows: 

“This study was supported by a grant to LB by the University of Rochester CTSA (UL1 TR000042) from the National Center for Advancing Translational Sciences of the National Institutes of Health. (https://www.urmc.rochester.edu/clinical-translational-science-institute.aspx).The funders had no role in study design, data collection and analysis, decision to publish, or preparation of the manuscript.”

I have submitted updated figure files, after using the PACE tool. 

We have considered submitting the protocol to protocols.io, but that platforms seems more conducive to laboratory protocols which this manuscript is not. For this reason, we have decided not to submit this protocol to protocols.io at this time.

Journal requirements response: 

1. I have reviewed the PLOS ONE’s style requirements, including those for the name filing. and believe that this manuscript complies with the PLOS ONE style templates provided. 

2. The updated financial disclosure statement is above, on this page.

3. I have removed the Acknowledgements section of the manuscript, as the funding information did not belong there. The funding information has been removed from the text in the manuscript.

4. Regarding Data Availability, this manuscript presents a study protocol, the research is actively ongoing, and data collection and analysis are not yet complete. Consequently, there are no final datasets available to share at this stage.

5. I have reviewed the reference list to make sure it is correct, and I have not found any of the articles cited to have been retracted. One additional reference has been added to list.

6. Regarding abbreviations, I removed PD and changed it to Positive Deviance, I removed SHRL and changed it to sexual and reproductive health literacy, I removed SRHC and changed it to sexual and reproductive healthcare and I removed CAC replaced it with Community Advisory Committee. For the acronym ‘RMAYW,’ because this is lengthy and written so many times throughout the manuscript, I left the acronym in most places, but I replaced it with the full words in the discussion section.

7. I have reviewed my reference list and believe it is complete and correct. I have added on reference.

In the “Reviewer Responses to Questions” section, the reviewers note that the manuscript does not provide the information required by the PLOS Data policy. Again, with regard to the questions about data Regarding Data Availability, this manuscript presents a study protocol, and the research is actively ongoing, and data collection and analysis are not yet complete. Consequently, there are no final datasets available to share at this stage.

Reviewer #1

Thank you for acknowledging to our efforts to conduct research with marginalized populations. We appreciate your review and your important and useful critique that has resulted in an improved manuscript. 

Reviewer #1 

Every research endeavor demands a significant investment of effort; therefore, maximizing its validity whenever possible is essential. Two key suggestions to enhance validity include referencing the (1) mixed-method research protocol outlined by Venkatesh, V., Brown, S.A., and Sullivan, Y.W. (2016) in the "Guidelines for Conducting Mixed-methods Research: An Extension and Illustration," Journal of the AIS (17:7), 435-495, and (2) utilizing the forward-backward translation technique proposed by Guillemin F, Bombardier C, and Beaton D. in "Crosscultural adaptation of health-related quality of life measures: literature review and proposed guidelines," J Clin Epidemiol. 1993;46:1417-1432.

 In an effort to maximize validity of the mixed methods protocol in this manuscript, when applicable, I have used Venkatesh et al., 2016’s decision tree to add detail in several places on the manuscript. See lines 174-177, 279, 520, The reference has been added to the reference list.

I have added details about the translation technique that we used for this protocol. In the future I will consider following the direction of the reference you suggested. (Lines 479-482)

1. In lines 36-38, to clarify the role of the Community Advisory Committee (CAC) in this study, it may be beneficial to explicitly state that the CAC will be formed to provide community-engaged guidance and support for the recruitment of participants. I added your suggestion.

2. In line 87, does "successful" refer to the criteria detailed in lines 212-215? If so, it would enhance clarity for readers to provide examples of what is meant by "success" immediately. This is important as "success" is one of your main definitions. I added a description of ‘successful,’ lines 91-92.

3. Line 90-94: Are these two sentences the same?

 Yes, essentially. I deleted one of them.

4. In line 128-132, according to Figure 1, "environmental" is one of the variables, but it is omitted in Aim 1. Is this intentional, and if so, why?

 It was unintentionally omitted; I have since added it.

5. In line 143, to ensure the validity of the mixed method protocol, please consider incorporating the mixed-method protocol suggested by Venkatesh, V., Brown, S.A., and Sullivan, Y.W. (2016) in the "Guidelines for Conducting Mixed-methods Research: An Extension and Illustration," Journal of the AIS (17:7), 435-495. For instance, questions regarding the property of mixed-method research, such as whether quantitative and qualitative data collection occur sequentially or concurrently, are not explicitly addressed.

 I’ll repeat a change described above… to maximize validity of the mixed methods protocol in this manuscript, when applicable, I have used Venkatesh et al., 2016’s decision tree to add detail in several places on the manuscript. See lines 174-177, 279, 520. The reference has been added to the reference list.

6. In line 161-164, according to Figure 1, "environmental" is one of the variables, but it is omitted in Aim 1. Is this intentional, and if so, why?

 It had been unintentionally left off, but I have since deleted this sentence due to redundancy (as requested by another reviewer).

7. In line 188, are "medical providers" and "healthcare providers" referring to the same group of individuals? This question arises because in Figure 2, "medical providers" are not included. Yes, I have updated the wording to say ‘healthcare providers’ for consistency. Thank you.

8. In line 264, could you please clarify what occurs if participants choose to withdraw after completing the interview?

Line 269: What is HIPAA? Do you mean the Health Insurance Portability and Accountability Act? Please clarify the full term for readers who may not be familiar with the abbreviation.

 I added to this line to say that if they choose to withdraw at the end of the interview, they may choose to have their interview and or survey redacted.

Yes, it is and I removed the acronym and put the full name there.

9. Line 295: Is it 5 or 6 options? Please double-check. Has been updated to read: “…six optional answers’

10. In line 302, why are only 3 questions out of the 5 items from the Centrality of Religiosity Scale (CRS-5) being used to measure religiosity? Please clarify

 I have added the following explanation: 

Two questions were omitted from the 5-question scale based on preliminary feedback stating that two questions appeared too cumbersome, intrusive, and unnecessary for this population.

11. In line 307, could you explain how the validity of the translation is ensured, particularly considering the utilization of forward and back translation techniques?

 I see the value in the forward and back translation, but have added the following statement as this is what actually took place in the study: 

The principal investigator will initially translate the questions from English to Spanish, and then the bilingual Community Advisory Committee members will review the translation for accuracy and modify as needed.

12. In lines 328 and 329, please consider including forward and back translation techniques to ensure validity.

Line 364: "RYMAW," is this a typo?

 The principal investigator will initially translate the questions from English to Spanish and then the bilingual CAC members will review for accuracy of the translation and modify as needed.

Yes, it was a typo, and it has been fixed.

13. Please clarify if the data collected would be share with the journal or not With regard to the questions about data availability, this manuscript presents a study protocol, and the research is actively ongoing, data collection and analysis are not yet complete. Consequently, there are no final datasets available to share at this stage.

14. Please check the labeling of Fig 2. I have relabeled Fig 2.

Reviewer #2

Thank you for taking the time to review our manuscript. I appreciate your suggestions to improve clarity to enhance consistency of application of this protocol. 

Reviewer #2 

1. Introduction section, 3rd paragraph, 3rd to last and 2nd to last sentences: It was not clear what distinct information is being conveyed by these two sentences (one starting with “While barriers and facilitators to healthcare …” and the other with “While barriers and facilitators to care …”). Please consider either combining them or further differentiating their content from one another.

 Thanks, I deleted one of these sentences.

2. Near the end of the Introduction section, where the three specific aims are mentioned, please consider sharpening the distinction between Specific Aims 2 and 3.

 Aim 2: I added ‘they have’ to indicate I will be identifying strategies that they have used to gain health literacy and healthcare access. 

Aim 3: I will be using participants’ recommendations or suggestions (which are different than the strategies they have used) that they have to improve health literacy and healthcare access and collaborate with the CAC to form recommendations to share at the end of the study.

3. Throughout the Methods section, please consider making clearer the exact study tasks to be carried out for each specific aim.

 I have added information on 496-497 & 524-530 linking data analysis with the specific aims.

4. The Study Design section’s 5th sentence (which starts with “National data indicate low health literacy … for these women …”) seems to contradict the Introduction section’s 3rd paragraph’s 3rd sentence, which states that “statistics regarding the health literacy of this specific group are unavailable.” Please consider resolving this seeming discrepancy.

 Thank you, based on another reviewer’s feedback I added that information to the introduction part and deleted it from there. It may have seemed contradictory, in an effort to clarify, I can say that there are more statistics about Hispanic women compared with Mexican American women, but the few studies that are of Mexican American women indicate lower health literacy in this group. (I agree this could have been more clearly stated originally).

5. The Study Participants and Eligibility section’s 3rd sentence states that “precautions will be taken not to obtain any personal identifying information,” while the Eligibility Screening and Enrollment section’s last sentence mentions the storage of identifying information. Please consider clarifying how these planned tasks are consistent.

 I added a clarifying sentence to the Eligibility screening and enrollment section: 

If eligibility screening takes place in person and if the interview is to be conducted at that time, contact information will be unnecessary and will not be collected. If eligibility screening occurs on REDCap, contact information (name, email and phone number) will be collected in order to arrange a meeting time for an interview.

6. In the Recruitment section, please consider elaborating on whether the study information disseminated through the flyers and posters will be made available to potential participants electronically/virtually as well (and if so, how), in addition to being displayed and distributed at clinic locations and extra flyers being available for potential participants to physically pick up. I have added to lines 376-378 to describe how the flyers may be electronically disseminated. 

7. Data Collection section: Please consider providing a justification for why the extra layer of protection using the voice changing platform is by choice of the participant and only offered to some participants, rather than being uniformly applied to all participants.

 The voice altering option was implemented in response to the IRB’s concern of risk to participants who may be undocumented. 

The sentence has been altered and is as follows: 

To minimize risk, an optional voice changing platform will be offered, as an additional layer of protection to participants, as a consideration for those who may be undocumented immigrants. In this case, the original audio recording of the interview will be re-recorded using the voice changing platform and stored and used for analysis in the altered form.

8. Regarding data availability, this submission is accompanied by the statement “All relevant data from this study will be made available upon study completion.” It is unclear how this will be done and what steps will be taken to protect participant confidentiality in doing so, especially given the potentially sensitive topics covered by the semi-structured interviews.

 Given that this is a protocol paper, 

the research is actively ongoing, and data collection and analysis are not yet complete. Consequently, there are no final datasets available to share at this stage.

While revising your submission, please upload your figure files to the Preflight Analysis and Conversion Engine (PACE) digital diagnostic tool. I have used the PACE tool to update my figure files.

Editor (Mary D. Clark) 

Line 82 starting with in general to 83 (8) is really repetitive can you take it out?

This what it was:

In general, Hispanic individuals are disproportionally affected by poorer health literacy [8], which is exacerbated for those who were raised speaking Spanish or who are monolingual Spanish speakers [9], as well as for those residing in rural areas

 Here is the updated sentence:

Health literacy is lower for those who were raised speaking Spanish or who are monolingual Spanish speakers.

Description of edits made I changed Aim 1 to two sentences to decrease the length of that sentence.

Description of edits made Line 154-156 – deleted and added the pap screening to introduction, line 78-79

Redundancy reduction: 

The study will explore each of the core domains that comprise health literacy including knowledge, competence, motivation, access, understanding, appraisal, and application of health information as well as societal, situational, and personal determinants [22] as they relate to successful access of SRHC services. 

 Deleted lines 161- 164 (this entire sentence)

Deleted the following to reduce redundancy:

The purpose of the CAC will be to provide distinct perspectives from those who provide care as well as community members who have a longer view of the women’s sexual and reproductive health experiences in their communities. 

 Deleted lines 176-17

---

## [Editor Report · Decision Letter 1]

6 May 2024

Study protocol: Examining sexual and reproductive health literacy in Mexican American young women using a positive deviance approach

PONE-D-23-42586R1

Dear Dr. Batek,

We’re pleased to inform you that your manuscript has been judged scientifically suitable for publication and will be formally accepted for publication once it meets all outstanding technical requirements.

Kind regards,

Mary Diane Clark, PhD

Academic Editor

PLOS ONE

Additional Editor Comments (optional):

Thank you for the careful revisions. The paper reads much better and with fewer abbreviations it is easier to access. Nice work
---

## [Editor Report · Acceptance letter]

13 May 2024

PONE-D-23-42586R1 

PLOS ONE

Dear Dr. Batek, 

I'm pleased to inform you that your manuscript has been deemed suitable for publication in PLOS ONE. Congratulations! Your manuscript is now being handed over to our production team.

Kind regards, 

on behalf of

Dr. Mary Diane Clark 

Academic Editor

PLOS ONE